# Novel Amphiphilic Cyclobutene and Cyclobutane *cis*-C_18_ Fatty Acid Derivatives Inhibit *Mycobacterium avium* subsp. *paratuberculosis* Growth

**DOI:** 10.3390/vetsci6020046

**Published:** 2019-05-24

**Authors:** Denise K. Zinniel, Wantanee Sittiwong, Darrell D. Marshall, Govardhan Rathnaiah, Isin T. Sakallioglu, Robert Powers, Patrick H. Dussault, Raúl G. Barletta

**Affiliations:** 1School of Veterinary Medicine and Biomedical Sciences, University of Nebraska–Lincoln, Lincoln, NE 68583, USA; dzinniel2@unl.edu; 2Department of Chemistry, Faculty of Science and Technology, Thammasat University, Rangsit Campus, Phaholyothin Rd., Klong Luang, Pathum Thani 12121, Thailand; wantane@tu.ac.th; 3Total Analysis Limited Liability Company, Detroit, MI 48204, USA; ddmarshall402@gmail.com; 4Materials and Machines Corporation (MatMaCorp), Lincoln, NE 68507, USA; grathnaiah@matmacorp.com; 5Department of Chemistry, University of Nebraska–Lincoln, Lincoln, NE 68588, USA; isin-tuna.sakallioglu@huskers.unl.edu (I.T.S.); rpowers3@unl.edu (R.P.); patrick.dussault@unl.edu (P.H.D.); 6Nebraska Center for Integrated Biomolecular Communication, University of Nebraska-Lincoln, Lincoln, NE 68588, USA

**Keywords:** *cis*-C_18_ fatty acids, cyclobutane derivatives, cyclobutene derivatives, growth inhibition, *Mycobacterium avium* subsp. *paratuberculosis*

## Abstract

*Mycobacterium avium* subspecies *paratuberculosis* (Map) is the etiologic agent of Johne’s disease in ruminants and has been associated with Crohn’s disease in humans. An effective control of Map by either vaccines or chemoprophylaxis is a paramount need for veterinary and possibly human medicine. Given the importance of fatty acids in the biosynthesis of mycolic acids and the mycobacterial cell wall, we tested novel amphiphilic C_10_ and C_18_ cyclobutene and cyclobutane fatty acid derivatives for Map inhibition. Microdilution minimal inhibitory concentrations (MIC) with 5 or 7 week endpoints were measured in Middlebrook 7H9 base broth media. We compared the Map MIC results with those obtained previously with *Mycobacterium tuberculosis* and *Mycobacterium smegmatis*. Several of the C_18_ compounds showed moderate efficacy (MICs 392 to 824 µM) against Map, while a higher level of inhibition (MICs 6 to 82 µM) was observed for *M. tuberculosis* for select analogs from both the C_10_ and C_18_ groups. For most of these analogs tested in *M. smegmatis*, their efficacy decreased in the presence of bovine or human serum albumin. Compound 5 (OA-CB, 1-(octanoic acid-8-yl)-2-octylcyclobutene) was identified as the best chemical lead against Map, which suggests derivatives with better pharmacodynamics may be of interest for evaluation in animal models.

## 1. Introduction

Johne’s disease (JD) is caused by *Mycobacterium avium* subspecies *paratuberculosis* (Map), and is characterized by enteritis that affects ruminant species and wildlife. The disease has a global distribution and is a significant problem for the livestock and associated industries [1]. Recent reports have suggested a link to Crohn’s disease, raising concerns that Map is a zoonotic and/or food-borne pathogen [2]. As a result, identification of effective methods for control of JD is important in terms of trade, public health, and national agrosecurity [3]. Although an effective control of JD remains a major challenge, control strategies are continuously being revised, including the use of vaccines and therapeutic drugs for treatment and/or chemoprophylaxis [4]. In the context of JD, vaccines and chemoprophylactic approaches are suitable for use in the field, while treatment options are primarily considered for high-value breeding stock animals. If Map is eventually demonstrated to cause at least a subset of Crohn’s disease cases, Map chemotherapeutics could become important human medicinal agents.

Although standard drugs used for treatment of *Mycobacterium tuberculosis* (Mtb) do not have the same efficacy on microorganisms of the *Mycobacterium avium* group, antimycobacterial agents such as isoniazid (Inh) and clofazimine have been used, on an extra-label basis, to treat JD in animals of significant economic value [5,6]. Monensin, a polyether antibiotic, has also been used for prophylaxis in calves and to reduce fecal shedding in Map-infected animals [7]. However, the number of chemotherapeutic agents that can be used either for prophylaxis or treatment remains limited.

There is a long history on the development of fatty acids and related compounds as potential chemotherapeutic agents against mycobacteria [8,9]. In previous studies, we reported the chemical synthesis of a novel series of amphiphilic cyclobutene and cyclobutane fatty acid derivatives possessing substantial in vitro efficacy against Mtb and limited off-target toxicity [10]. Mycobacteria are known to directly incorporate fatty acids into mycolic acid biosynthesis [11], and a number of classical mycobacterial drugs, including Inh and ethionamide, target this essential process [12]. The presence of mycolic acids is a general feature of mycobacteria; however, different mycobacterial species have specific structures that could affect the efficacy of various fatty acid derivatives [13]. Our hypothesis is that uptake of the modified fatty acids results in the inhibition of mycolic acid biosynthesis by an uncharacterized mechanism. Thus, those inhibition-associated alterations in the cell wall structure may lead to lethality, increased susceptibility to antibiotics or the host immune system, or decreased pathogen virulence.

Cyclobutanes can be found within the structures of a large number of natural products and the backbone of the clinically used anticancer drug carboplatin [14,15,16,17]. Although the enhanced reactivity of cyclobutenes and cyclobutanes compared to unstrained cycloalkanes makes both groups valuable as synthetic intermediates [18], neither functionality is particularly fragile. Our previous work investigated two series of highly specific fatty acid derivatives, one based on decenoic acid (C_10_) and the other on oleic/elaidic acids (C_18_) [10]. We also demonstrated that these carbocylic analogs withstood routine synthetic processing and brief heating to temperatures of at least 100 °C [19]. We found active compounds from each series that had lower minimal inhibitory concentrations (MICs) against Mtb on a molar basis compared to Inh. In this manuscript, we report the inhibitory properties of fatty acid compounds on Map.

## 2. Materials and Methods

### 2.1. Bacterial Strains and Culture Conditions

Wild-type strains utilized in this study were the original virulent clinical isolate of Map K-10 [20] and *Mycobacterium smegmatis* (Msm) mc^2^155 [21]. Map cultures were grown standing at 37 °C in complete Difco™ Middlebrook 7H9 base broth (Becton, Dickinson and Company; Sparks, MD, USA), adjusted to pH 5.9, and supplemented with oleic acid/albumin/dextrose complex (BBL™; Becton Dickinson Microbiology Systems, Franklin Lakes, NJ, USA), ferric mycobactin J (Allied Monitor; Fayette, MO, USA) and Tween^®^ 80, as previously described [20]. Msm cultures were grown at 37 °C with shaking (200 rpm; Innova 4300 incubator shaker; New Brunswick Scientific; Edison, NJ, USA) in complete 7H9 broth supplemented with albumin/dextrose complex and Tween^®^ 80, as described [22]. Cultures were grown to an OD_600_ between 0.6 and 1.2.

### 2.2. Fatty Acid Derivatives

The fatty acid derivatives, illustrated in Table 1 and Figure 1, were prepared using straightforward procedures. Depending upon the target structure, between three and six synthetic steps were required to synthesize these compounds in the laboratory [10], which can readily be scaled-up for drug industry development. 1D ^1^H NMR spectroscopy was used to verify the chemical purity of these fatty acids. Four of the molecules (DA-CB, 1; DA-satCB, 2; OA-CB, 5; and OA-satCB, 6) are single diastereomeric compounds, while DA-alcCB (4) is a mixture of two diastereomers differing in the stereochemistry of the hydroxyl (OH) group. Derivatives OA-alcCB (8), OA-ketoCB (7), and EA-ClketoCB (11) are mixtures of regioisomers in terms of the position of the substituents on the ring relative to the “head” and “tail” of the fatty acid structure; the latter two are each a mixture of two diastereomers. All of the compounds are racemic, except for DA-satCB (2), which is achiral. The antituberculosis drug D-cycloserine (DCS; Sigma Life Science; St. Louis, MO, USA) was used for comparative purposes.

### 2.3. Drug Susceptibility Assays

MICs were determined by the 96-well microtiter plate method with three technical and three biological replicates, as previously described [22]. Once the bacteria reached the proper OD_600_, cells were washed twice in 7H9 base broth, 25 mM/L glycerol, and 0.02% v/v tyloxapol. For Map, 2.0 mg/L ferric mycobactin J was also added. Cells were vortexed for one minute with 4.0 mm glass beads to disrupt any clumps that may have formed during the centrifugation and washing steps. Map cells were inoculated at an initial concentration of ca. 2.5 × 10^6^ or 2.5 × 10^5^ colony forming units (CFU)/well, and the microtiter plates were incubated at 37 °C for 5 or 7 weeks, respectively. Msm cells were inoculated at ca. 1.0 × 10^5^ CFU/ml, recording results after 4 days at 37 °C, as described [10]. The initial inocula were plated to quantify retrospectively CFU/ml for each strain and level.

We tested eight fatty acid derivatives against Map and seven against Msm, as well as decenoic acid (DA, 12); in all cases, the compounds were prepared as solutions in 100% DMSO-d_6_ (Sigma Life Science). For Map, the doubling dilution gradients were prepared from 16 to 1024 µg/ml, adjusting the concentration of DMSO-d_6_ in all wells to 1.024% v/v. For Msm, the doubling dilution gradients were prepared from 32 to 2048 µg/ml, which corresponded to 2.048% v/v of DMSO-d_6_. MIC differences for a given compound are considered significant when values differ by at least 2 doubling dilutions. At these concentrations, DMSO-d_6_ had no growth inhibitory effect on either Map or Msm cells. In addition, the potential interference of serum albumins in MIC assays against Msm was analyzed in a separate experiment by supplementing the media with various concentrations of Fraction V bovine serum albumin (BSA; Fisher Scientific Fair Lawn, NJ, USA) or human serum albumin (HSA; Sigma Life Science).

## 3. Results

### 3.1. Inhibitory Activity of Fatty Acid Derivatives against Msm

As Map does not grow well in the defined minimal media used in our previous studies [10], we first compared the susceptibility of a control strain, Msm mc^2^155, in Middlebrook 7H9 base broth. This broth contains, in addition to the minimal media components, L-glutamic acid, sodium citrate, pyridoxine, biotin, and copper sulfate. As complete Middlebrook 7H9 contains albumin, we also compared the effects of supplementing additional amounts of albumin to the base broth. Establishing MIC values in the presence of BSA and/or HSA, conditions relevant to mammalian serum, is important in interpreting in vitro and in vivo compound efficacy data for potential applications to veterinary or human medicine. DCS, an effective second-line antituberculosis drug, was employed as a point of reference against the unknown inhibitors.

Experimental results are presented in Table 2. The MICs measured in the presence of increasing levels of BSA or HSA fell within a narrow concentration range. Moreover, MICs for the majority of the analogs differed at most by one doubling dilution compared to values previously obtained in minimal media without albumin (Table 2, Columns 2, 3, and 7). These differences are within the experimental error of this assay as indicated by the two replicates, all measured at day 4 post-inoculation in the same media but performed at different times (Table 2, Columns 3 and 7). In contrast, the MICs of OA-CB (5) and EA-ClketoCB (11) were significantly different in Middlebrook 7H9 base broth and minimal media.

Albumins, which are known to strongly bind fatty acids [23,24], are expected to protect the microorganisms from drug activity, with concomitant decrease in susceptibilities and increased MICs. We tested both BSA (Table 2, Columns 4–6) and HSA (Table 2, Columns 8–10) at 0.125, 0.25, and 0.5% w/v. The C_10_ compound (DA-CB, 1), the control C_10_ decenoic acid parent compound (DA, 12), and DCS displayed the same MICs or values within one doubling dilution for both BSA and HSA at all concentrations tested. Similar behavior was also observed for DA-satCB (2) and OA-CB (5) with HSA and BSA, respectively. Compounds OA-satCB (6), OA-ketoCB (7), OA-alcCB (8), and EA-ClketoCB (11) displayed significant differences in MICs in the range of two to three doubling dilutions for both BSA and HSA. In addition, the MICs for fatty acid derivatives DA-satCB (2) and OA-CB (5) varied by two or more doubling dilutions for BSA and HSA, respectively. In general, the effect of albumins reached a saturation point at 0.25% without further changes in MICs. 

### 3.2. Inhibitory Activity of Fatty Acid Derivatives against Map

To test Map susceptibility, we supplemented the 7H9 base broth with ferric mycobactin J, as described in Section 2.3. The results for both 5 and 7 week endpoints (Table 3, Columns 2 and 3) indicated no significant differences since, at most, MICs differed by one doubling dilution. Specifically, the longer-chain “OA” compounds OA-CB (5), OA-ketoCB (7), and OA-alcCB (8) were active with MICs of ca. 400 µM. The “DA” shorter-chain fatty acid derivatives DA-CB (1) and DA-satCB (2) were less active with MICs of ca. 1300 µM or higher. Intermediate MIC activity of 713 µM was seen for the analog based upon the elaidic acid framework (EA-ClketoCB, 11). DCS displayed activity similar to the less effective compounds with MICs of 2508 and 1254 µM for weeks 5 and 7, respectively.

As indicated above in Table 2 for the Msm mc^2^155 wild-type, the differences in media compositions were minor and did not affect the MICs. Thus, we also compared the Map week 7 results with those previously obtained with Mtb [10]. In general, Mtb was more susceptible to the fatty acid derivatives and DCS than Map. For example, while OA-alcCB (8) was one of the best inhibitors for both Map and Mtb, MICs against Map were 16- to 65-fold greater than than those for Mtb strains CDC1551 and H37Rv, respectively. In terms of relative efficacy within the series of analogs, DA-CB (1), DA-satCB (2), OA-satCB (6), and OA-alcCB (8) have MICs < 100 µM against Mtb. Fatty acid derivatives OA-CB (5), OA-ketoCB (7), and OA-alcCB (8) displayed MICs < 500 µM against Map. Compounds DA-satCB (2), OA-satCB (6), and OA-alcCB (8) proved to have greater activity than DCS towards Mtb; the same was true in Map for OA-CB (5), OA-satCB (6), OA-ketoCB (7), OA-alcCB (8), and EA-ClketoCB (11). Fatty acid derivative EA-ClketoCB (11), while exhibiting identical MICs (713 µM) for Map and Mtb, is one of the least effective Mtb inhibitors, but is still a moderate inhibitor of Map. Likewise, DA-satCB (2) was highly effective against both Mtb strains, but had a lower efficacy against Map. 

## 4. Discussion

This study demonstrated that fatty acid analogs previously shown to inhibit Mtb growth also inhibited Map growth, except compound 4 (DA-alcCB), which had an MIC > 4778 µM. However, there are major differences in the MICs and the compound structures that were more active against each species. The four most active fatty acid derivatives tested against Mtb had MICs in the range of 20–82 µM. In contrast, the three analogs displaying the greatest activity against Map have MICs of 392–415 µM. The control compound (DA, 12) was moderately active (MIC 188 µM) against Mtb, but this compound was essentially inactive against Map with MICs above 3000 µM. The one exception to this trend is chloroketone (EA-ClketoCB, 11), which has the same MIC (713 µM) against Mtb and Map. Our investigations with Mtb found good inhibitory activity for analogs in both the C_10_ and C_18_ series; whereas, the analogs with the lowest MICs against Map came exclusively from the C_18_ series derived from oleic acid. Three of the four analogs most active against both Mtb strains (MICs < 100 µM) are based upon a simple cyclobutene (DA-CB, 1) or cyclobutane (DA-satCB, 2; OA-satCB, 6) structure. Of note, the C_18_ oleic acid parent compound is not inhibitory for Map since oleic acid is an enrichment recommended for complete Middlebrook media. Regarding the three analogs most active against Map, OA-CB (5) is a simple cyclobutene, while the other two contain oxygen-functionalized cyclobutanes (OA-ketoCB, 7; and OA-alcCB, 8). Two analogs screened against *E. coli* G58-1—OA-satCB (6) and DA-CB (1)—had weak inhibition at concentrations up to 2608 µM [10]. Thus, these compounds may inhibit a specific metabolic pathway unique to mycobacteria.

In the earlier studies with Mtb [10], we observed that most analogs tested here were more active in minimal than in rich media (LB broth), suggesting that interactions of fatty acids with albumins or other media components could result in reduced efficacy. In our current studies, we tested the effects of albumins using Msm as a model system. BSA or HSA supplementation had the greatest effect on the efficacy of compounds 6 (OA-satCB), 7 (OA-ketoCB), 8 (OA-alcCB), and 11 (EA-ClketoCB). BSA or HSA supplementation increased the MIC by two or more doubling dilutions. An albumin-associated change was also observed for DA-satCB (2) and OA-CB (5). Despite the negative impact on MICs, the use of albumins may increase the solubility of these fatty acid analogs, which, in turn, may facilitate delivery into animals and improve pharmacokinetics, especially those properties related to absorption, distribution, metabolism, and excretion (ADME) [25]. These Msm results suggest albumins may have a similar impact on compound efficacy against Map and Mtb.

In evaluating which fatty acid analogs are the most promising leads to be developed into potential Map treatments, compound hydrophobicity and aggregation must also be considered [26,27,28]. Critical micellar concentrations (CMC) of four analogs in the C_18_ series (OA-satCB, 6; OA-ketoCB, 7; OA-alcCB, 8; and EA-ClketoCB, 11) have been established to be < 50 μM, while the CMC for compound 5 (OA-CB) was between 50 and 100 μM. It is important to note that the CMCs are well below the MICs identified here. The analogs in the C_10_ series (DA-CB, 1; DA-satCB, 2; and DA-alcCB, 4) have CMCs > 1000 μM, which may or may not encompass some of the highest concentrations employed here. It should be noted, however, that the CMCs were determined in a simple potassium phosphate alkaline buffer with DMSO-d6 [10], and the CMCs will likely increase upon the addition of fat-solubilizing albumins. The albumin-associated solubilization may be relevant for in vivo testing of these derivatives [29], especially for compounds whose MICs are least effected by albumins.

Fatty acid analogs investigated here, except compounds 7 (OA-ketoCB) and 11 (EA-ClketoCB), were compared against DA (12), Inh, and DCS for cytotoxicity against RAW 264.7 murine macrophages up to 250 µM, as shown in Table S1 of our previous manuscript [10]. In addition, the C_18_ oleic acid parent compound was shown to be nontoxic for this cell line (Ji-Young Lee and Patrick Dussault unpublished results). The analog concentration required to reduce cell viability below 50% proved to be structure-dependent: DA-CB (1, ca. 100 µM); DA-satCB (2, ca. 50 µM); DA-alcCB (4, > 250 µM); OA-CB (5, > 250 µM); OA-satCB (6, > 250 µM); OA-alcCB (8, < 25 µM); DA (12, ca. 200 µM); DCS (> 250 µM); and Inh (> 250 µM). Compounds 5 (OA-CB) and 6 (OA-satCB) were less toxic against murine macrophages, but displayed good to moderate activity against Map (Table 3). In contrast, fatty acid derivative 8 (OA-alcCB) had good inhibitory activity against Map, but was relatively toxic against RAW 264.7 cells. Moreover, supplementation with albumins may decrease fatty acid cytotoxicity; for example, the presence of BSA and HSA reduced apoptotic effects induced by long chain omega-3 polyunsaturated fatty acids in HaCaT epidermal keratinocyte cells [30]. Overall, compound 5 (OA-CB) was identified as the best chemical lead for further investigation. Chemical synthesis of derivatives of this compound with better pharmacodynamics may be of interest for further evaluation and testing in animal models.

We show that C_18_ fatty acid derivatives are better inhibitors of Map and, in general, both C_10_ and C_18_ analogs exhibited higher activity against Mtb than Map (Table 3). We predict that the mechanism of action for both C_10_ and C_18_ compounds is the ability to inhibit mycolic acid biosynthesis. In addition, membrane fluidity affects the compounds ability to penetrate further into the mycomembrane. The mycobacterial cell wall is comprised of mycolic acids that are essential high molecular weight α-alkyl, β-hydroxy fatty acids [11,31,32]. There are different structural classes of mycolic acids (α-, methoxy and keto-). The α-branch is saturated with typically 24 carbon atoms, while the meromycolate arm usually contains 60 carbons. There are also species-specific differences, especially in the long meromycolate arms. These characteristics affect membrane fluidity that is directly linked to the solubility of lipophilic compounds. The main properties affecting fluidity are the length of meromycolate carbon chains and the presence of conformational constraints in the form of cyclopropane units. The greater the number of *trans*-cyclopropanations in the α-mycolic acids, the higher the melting point and the lower its fluidity [31,33]. Regarding the species tested in this and our previous studies, the increasing order of fluidity is *M. avium* > Msm > Mtb. Though this may not be the only reason, increased fluidity may allow further penetration of C_10_ derivatives deep into the mycolic acid layer to potentially gain access to the cytoplasmic membrane and the cytosol to inhibit key components of the mycolic acid biosynthetic and/or transport machinery. The less fluid Msm and *M. avium* layers may allow for better penetration of the more lipophilic C_18_ derivatives.

Differences in lipid metabolism between Msm, Map, and Mtb may also explain the greater susceptibility of Mtb to these analogs. In Mtb, there is an expansion of genes involved in lipid biosynthesis, while genes related to lipid degradation predominate in Msm and Map that have an intermediate ratio of lipid biosynthetic and degradation genes [34]. Thus, the inhibition of lipid biosynthesis in Mtb may be more critical and cause greater overall metabolic disruption. This resembles the effect of Inh that has also a greater inhibitory effect on Mtb than on Msm or Map [35]. In this interpretation, the fatty acid derivatives target(s) could be either in the essential cytoplasmic steps prior to uptake by the conserved essential membrane trehalose monomycolate transporter MmpL3 or the inhibition of the transporter itself [32,36]. In addition, the orthologous proteins from Mtb and Map are not identical (e.g., 70% identity and 78% similarity for MmpL3) and may have different affinities for the fatty acid derivatives. Our compounds were rationally designed to inhibit cyclopropanation reactions that occur in the cytosol [10,32]. However, these enzymes are not likely to be potential targets since mycobacterial cyclopropanases (e.g., CmaA1, CmaA2, Mma2, and Pca), though important in vivo, are not essential in vitro [37]. Thus, the inhibition of these enzymes may not account for microbial growth inhibition. 

## 5. Conclusions

In this study, we demonstrated that several members of a class of cyclobutene- or cyclobutane-containing fatty acid derivatives based upon either C_10_ or C_18_ skeletons are inhibitory to Map. In previous studies, these analogs have also been shown to inhibit growth of Msm and Mtb. In contrast to the results with Mtb, where inhibition was observed with analogs from both the C_10_ and C_18_ series, only the C_18_ compounds displayed significant efficacy against Map. We also demonstrated in Msm that the presence of albumins increases the MIC for some of the compounds. However, albumins may also incease analog solubility, and the concomitant decrease in cytotoxicity might allow for higher drug concentrations in vivo. While the level of inhibition displayed by these analogs against Map is moderate, the results suggest that this family of fatty acid derivatives may be a useful starting point in search of improved Map inhibitors. While we do not yet have a definitive explanation as to why Mtb is more susceptible to these analogs than Map, the results described in this manuscript, and in particular, the very different levels of inhibition observed with the two microorganisms across a common set of inhibitors, may reveal insights into their mechanisms of action. In future experiments, we plan to use metabolomic, biochemical, and genetic methods to validate the involvement of mycolic acid biosynthesis in the mechanism of action of these compounds against mycobacteria and identify the lethal target(s) affected. We also plan to determine the uptake of the fatty acid derivatives to identify whether the inhibition occurs in the cytoplasmic steps or the transport process.

## 6. Patents

A pending patent application, “Amphiphilic Cyclobutenes and Cyclobutanes” (Publication Number: US 2015/0175519 A1; Publication Date: Jun. 25, 2015; United States Application Number: 14/403,763), describes amphiphilic compounds containing a cyclobutene or cyclobutane moiety. In some embodiments, the compounds are useful for treating infection by *Mycobacterium* such as Mtb. Cyclobutene-containing compounds are also useful as monomers in the preparation of amphiphilic polymers.

## Figures and Tables

**Figure 1 vetsci-06-00046-f001:**
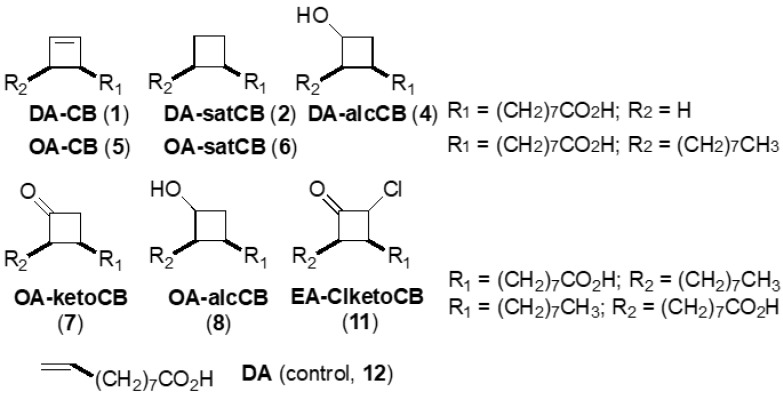
Structures of fatty acid derivatives investigated in this study and the decenoic acid control (DA, 12).

**Table 1 vetsci-06-00046-t001:** Fatty acid derivatives used in this study.

No.	Abbreviated Name ^2^	Compound Full Chemical Name	Formula Weight	Scaffold, Functionality
1	DA-CB	8-(2-Cyclobuten-1-yl)octanoic acid	196.3	C_10_, alkene
2	DA-satCB	8-Cyclobutyloctanoic acid	198.3	C_10_, alkane
4 ^1^	DA-alcCB	3-(Octanoic acid)cyclobutanol	214.3	C_10_, alcohol
5	OA-CB	1-(Octanoic acid-8-yl)-2-octylcyclobutene	308.5	*cis*-C_18_, alkene
6	OA-satCB	*cis*-9,10-Ethanooctadecenoic acid	310.5	*cis*-C_18_, alkane
7	OA-ketoCB	3-(Octanoic acid-8-yl)-2-octylcyclobutanone	324.5	*cis*-C_18_, ketone
8	OA-alcCB	3-(Octanoic acid-8-yl)-2-octylcyclobutanol	326.5	*cis*-C_18_, alcohol
11	EA-ClketoCB	2-Chloro-3-(octanoic acid-8-yl)-4-octylcyclobutanone	358.9	*trans*-C_18_, chloroketone
12	DA	Decenoic acid	170.2	C_10_, alkene (acyclic)
	DCS	D-cycloserine	102.1	

^1^ For the current study, this compound was only tested in Map. ^2^ CB = cyclobutene, satCB = saturated cyclobutene (cyclobutane), alcCB = cyclobutanol, OA = oleic acid, ketoCB = cyclobutanone, EA = elaidic acid, and ClketoCB = chlorocyclobutanone.

**Table 2 vetsci-06-00046-t002:** Minimal inhibitory concentration (MIC) values [µg/ml (µM)] against *Mycobacterium smegmatis* (Msm) with and without bovine serum albumin (BSA) and human serum albumin (HSA) at day 4.

No.	Minimal Media ^1^	No BSA ^2^	0.125% BSA	0.25% BSA	0.5% BSA	No HSA ^2^	0.125% HSA	0.25% HSA	0.5% HSA
1	512 (2608)	512 (2608)	512 (2608)	512 (2608)	512 (2608)	512 (2608)	1024 (5217)	1024 (5217)	1024 (5217)
2	512 (2582)	256 (1291)	512 (2582)	1024 (5164)	1024 (5164)	512 (2582)	512 (2582)	1024 (5164)	1024 (5164)
5	128 (415)	1024 (3319)	2048 (6639)	2048 (6639)	2048 (6639)	512 (1660)	1024 (3319)	2048 (6639)	>2048 (>6639)
6	256 (824)	512 (1649)	1024 (3298)	1024 (3298)	2048 (6596)	512 (1649)	2048 (6596)	2048 (6596)	2048 (6596)
7	128 (394)	256 (789)	512 (1578)	1024 (3156)	1024 (3156)	256 (789)	512 (1578)	1024 (3156)	1024 (3156)
8	256 (784)	256 (784)	512 (1568)	1024 (3136)	1024 (3136)	256 (784)	1024 (3136)	1024 (3136)	2048 (6272)
11	128 (357)	1024 (2853)	2048 (5706)	>2048 (>5706)	>2048 (>5706)	512 (1426)	1024 (2853)	2048 (5706)	2048 (5706)
12	512 (3007)	1024 (6015)	1024 (6015)	1024 (6015)	1024 (6015)	1024 (6015)	2048 (12029)	2048 (12029)	2048 (12029)
DCS	64 (627)	128 (1254)	128 (1254)	128 (1254)	128 (1254)	128 (1254)	128 (1254)	128 (1254)	128 (1254)

^1^ Data taken from [10]. ^2^ Data represents replicated experiments carried out at different times.

**Table 3 vetsci-06-00046-t003:** MICs [µg/ml (µM)] against *Mycobacterium tuberculosis* (Mtb) and *Mycobacterium avium* subspecies *paratuberculosis* (Map).

Compound #	Map K-10 Week 5	Map K-10 Week 7	Mtb CDC1551 ^1^ Week 7	Mtb H37Rv ^1^ Week 7
1	512 (2608)	256 (1304)	16 (82)	16 (82)
2	256 (1291)	256 (1291)	4 (20)	4 (20)
4	>1024 (>4778)	>1024 (>4778)	64 (299)	128 (597)
5	128 (415)	128 (415)	128 (415)	16 (52)
6	512 (1649)	256 (824)	8 (26)	16 (52)
7	128 (394)	128 (394)	64 (197)	32 (99)
8	128 (392)	128 (392)	8 (24)	2 (6)
11	256 (713)	256 (713)	256 (713)	256 (713)
12	1024 (6015)	512 (3007)	32 (188)	32 (188)
DCS	256 (2508)	128 (1254)	4 (39)	8 (78)

^1^ Data taken from [10].

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
