# Peer review of "Novel Amphiphilic Cyclobutene and Cyclobutane cis-C18 Fatty Acid Derivatives Inhibit Mycobacterium avium subsp. paratuberculosis Growth"

_vetsci, 2019, doi:10.3390/vetsci6020046_

Round 1
Reviewer 1 Report
The manuscript is well written. The manuscript described the fatty acid derivatives for the treatment of Johne’s disease (JD) caused by Mycobacterium avium subspecies paratuberculosis (Map). The synthetic compounds showed anti-bacteria activity probably due to inhibition of the mycolic acid biosynthetic and/or transport machinery. Thus, These compounds are expected to be promising agents for JD. Therefore, the manuscript is not too excellent to be published. In other words, the manuscript is so excellent that it should be published.
Comments
(1) Are synthesized 4-membered ringc fatty acids biologically and chemically stable?
(2) Is cyclobutene moiety synthesized easily? And is it appropriate for drug industry?
(3) The compounds in Table 3 exhibited higher activity against Mtb than Map. Why was it, simply?
(4) What was the inhibitory mechanism of the designed C18 compounds against Map?
(5) Future plan?
That is all.
Reviewer 2 Report
This is an interesting study and makes a useful follow-on (in Map) to the authors' previous work in Mtb. The hypothesis is a very logical one and one that has a great deal of appeal to workers in drug design and discovery in mycobacteria: the uptake of modified fatty acids can result in the inhibition of mycolic acid synthesis---by an as-yet uncharacterized mechanism---and this gives rise to lethality toward mycobacteria.
On line 255, the authors should punch up the summary. The study gives interesting new material, but it also raises useful new questions. A more thoughtful summary will set off the importance of this work.
Compounds were prepared as previously done in reference 10. Although one regrets to see that elemental analyses were not obtained for the compounds in that reference, satisfactory purity was indeed confirmed by other methods; so this is not the venue to quibble about that widespread recent trend away from elemental analysis in compound characterization.
